# Cascaded-Microrings Biosensors Fabricated on a Polymer Platform

**DOI:** 10.3390/s19010181

**Published:** 2019-01-06

**Authors:** Yuxin Liang, Qi Liu, Zhenlin Wu, Geert Morthier, Mingshan Zhao

**Affiliations:** 1School of Physics and Optoelectronic Engineering, Dalian University of Technology, Dalian 116023, China; liangyuxin0318@mail.dlut.edu.cn (Y.L.); zhenlinwu@dlut.edu.cn (Z.W.); 2Photonics Research Group, Department of Information and Technology, Ghent University, 9000 Ghent, Belgium; geert.morthier@ugent.be; 3Information Center, Shandong University, Jinan 250100, China; liuqi@sdu.edu.cn

**Keywords:** microring biosensor, polymer waveguide, Vernier effect, nanoimprint

## Abstract

Polymer-based single-microring biosensors usually have a small free spectral range (FSR) that hampers the tracing of the spectrum shifting in the measurement. A cascade of two microring resonators based on the Vernier effect, is applied in this article in order to make up for this defect. A small FSR difference between the reference microring and the sensing microring is designed, in order to superpose the periodic envelope signal onto the constituent peaks, which makes it possible to continuously track the spectrum of the sensor. The optical polymer material, Ormocore, which has a large transparent window, is used in the fabrication. The biosensor is fabricated by using an UV-based soft imprint technique, which is considered to be cost-effective and suitable for mass production. By optimizing the volume ratio of Ormocore and the maT thinner, the device can be fabricated almost without a residual layer. The device works at a wavelength of 840 nm, where water absorption loss is much lower than at the infrared wavelengths. A two-step fitting method, including single-peak fitting and whole-envelope fitting, is applied in order to trace the spectral shift accurately. Finally, the two-cascaded-microrings biosensor is characterized, and the obtained FSR is 4.6 nm, which is 16 times larger than the FSR of the single microring biosensor demonstrated in our previous work. Moreover, the sensitivity can also be amplified by 16-fold, thanks to the Vernier effect.

## 1. Introduction

Integrated photonics have been widely known as key components in optical communication [1,2,3,4]. In recent years, they have also attracted a lot of interest for biosensors, because of their unique features, such as high sensitivity, being label free, and allowing for real-time monitoring [5,6]. These advantages make them suitable for Lab-on-a-Chip (LOC), which can make disease detection easy, convenient, and possible without professional medical personnel. The evanescent tail of waveguide modes that can interact with the external environment is usually utilized in integrated photonics biosensors, and various structures have been demonstrated, such as Mach–Zehnder interferometers (MZIs) [7], Young interferometers [8], microring resonators [9,10], photonic crystals [11,12], and plasmonic structures [13,14,15]. Among them, the microring resonator is one of the most attractive structures, as its resonance allows for a high sensitivity. Microring biosensors have been demonstrated on different optical materials, e.g., SOI (Silicon-On-Insulator) [9], SiN (silicon nitride) [16], and polymers [17]. Polymers as optical materials are commonly considered as suitable for biosensing, because of both the requirements of the biosensors and the unique features of the polymers. Firstly, compared to inorganic materials, polymers have superior biocompatibility, which can simplify the waveguide surface treatment process. Secondly, specific receivers must be bonded onto the waveguide surface, in order to catch antibodies (or antigens) of specific diseases. However, after detection, the specific antibodies (or antigens) are hard to remove, and this means that recycling is difficult to realize. Therefore, the economy factor should be taken into consideration. Polymers are an ideal material option, as they are cost-effective, not only in terms of the material itself, but also in the fabrication process. Various novel fabrication processes have been reported, such as photolithography, electron beam lithography, and excimer laser micromachining [18,19]. At last, the polymer is usually transparent over a large wavelength window, including both the infrared wavelengths (1300–1550 nm) and the visible wavelengths. The transparency at short wavelengths makes it preferable for optical biosensor devices, as the absorption loss of water at short wavelengths is much lower than at infrared wavelengths [20]. Motivated by these reasons, a lot of work has been devoted to polymer-based microring biosensors [10,17]. However, the low refractive index contrast makes the bend loss relatively large, and a small roundtrip length of the microring cannot be reached. This means a small free spectral range (FSR) is inevitable, which makes the tracing of the resonance wavelength shift much more difficult in the measurement.

The Vernier effect, based on cascaded microrings, has been widely recognized in integrated optics applications, such as optical filters [21] and tunable lasers [22]. Recently, the structure has also been demonstrated for sensor applications. The Vernier effect is exploited by designing a slight difference between the FSRs of the two microrings (the sensing microring and the reference microring), and it can lead to a higher sensitivity and a larger FSR, compared to the single microring resonator. The structure was first proposed to realize the digital optical sensor, which exhibits isolated peaks [23]. This results in a discrete limit-of-detection (LOD), which is equal to the difference of the FSRs. In article [24], the structure is improved by designing a FSR difference between two microrings, which is small, compared to the full-width at half-maximum of the resonance peaks of the individual resonators. As a result, a periodic envelope signal can be superposed onto the constituent transmission peaks. By appropriate fitting, the resonance shift can be traced accurately. However, a small FSR difference also results in a large envelope period. In order to ensure that the envelope period falls within the wavelength range of the measurement devices, small FSRs for both microrings are required. This means that a large roundtrip by the microring is needed, which is not suitable for the SOI or SiN platforms. By contrast, it is much more suitable for polymer microrings, as the small FSR is its intrinsic feature. Moreover, the enhanced FSR by the Vernier effect can make up for the small FSR of the single polymer-based microrings. In addition, the sensitivity can be improved by the Vernier effect.

Motivated by the reasons described above, a biosensor based on two cascaded microrings is designed and fabricated onto the polymer platform. A small FSR difference between the reference microring and the sensing microring is designed to generate the Vernier effect, in order to improve the FSR and the sensitivity. Ormocore (Microresist Technology) as a commercial optical material, is adopted, which has a large transparent window. An improved version of the nanoimprint technique, the UV-based soft imprint technique (Soft UV NIL), is applied to fabricate the biosensor. It is usually considered to be simple and suitable for mass production, as no high pressures or temperatures are necessary, and roller-to-roller can be applied [25].

## 2. Theoretical Analysis and Design

The Vernier scale is well known as an approach to improve the accuracy of the measurement instrument, by using the interpolation method. Two scales, including a main scale and a subsidiary scale, are required to generate the Vernier effect. The measurement is performed by observing which of the subsidiary scale grades is co-incident with a grade on the main scale. Such an arrangement can obtain higher resolution by using higher scale ratios. It is normally used in calipers and micrometers. Nowadays, it also finds applications in integrated photonics biosensing, which utilizes two microring resonators as the main and subsidiary scales.

The schematic configuration of the biosensor is shown in Figure 1a. The cross-section and the propagating mode of the polymer-based waveguide are also shown in Figure 1b,c, respectively. Two microring resonators, including the reference microring and the sensing microring, are cascaded via a common bus waveguide. The output light of the drop port of the reference microring resonator is launched into the sensing microring resonator by the bus waveguide. Each microring resonator has a comb-like transmission spectrum, which is illustrated in Figure 2a. The solid line shows the spectrum of the sensing microring resonator, while the dashed line shows the spectrum of the reference microring resonator. In order to generate the Vernier effect, a slight difference between the FSRs of the two microring resonators is designed, by applying different radiuses. The sensing principle can be described, as below. As shown in Figure 2a, both resonators have the same resonance wavelength (841.5 nm), and the highest peak of the whole device spectrum appears at this wavelength, which is highlighted in Figure 2b. It also can be seen that the neighboring peaks are inhibited, due to resonance wavelength differences between the two resonators. The reference microring resonator is usually covered by an upper cladding in order to prevent the influence from the external environment. In contrast, the sensing microring resonator should be exposed to the analyte solution, and the resonance wavelength shifts with the variation of the refractive index of the analyte solution. It can be observed from Figure 2 that when the resonance wavelength of the sensing microring resonator shifts by Δλ which is equal to the FSR difference, the highest peak hops to the next resonance wavelength of the reference microring resonator. In other words, a Δλ wavelength shift of the sensing microring spectrum will result in a wavelength shift of the whole device spectrum by ${\mathrm{FSR}}_{r}$. This means that the sensitivity can be amplified M times, where M can be expressed as [26]:(1)$$M=\frac{{\mathrm{FSR}}_{r}}{\left|{\mathrm{FSR}}_{r}-{\mathrm{FSR}}_{s}\right|}\;$$
where ${\;\mathrm{FSR}}_{r}\;$ and ${\mathrm{FSR}}_{s}\;$ represent the FSR  of the reference microring and the sensing microring, respectively. It can be deduced that the FSR of the total spectrum (${\mathrm{FSR}}_{t})$ will also be amplified M times compared to ${\mathrm{FSR}}_{s}$ as shown in Figure 2c. The ${\mathrm{FSR}}_{t}\;$can be expressed as:(2)$$FSR_{t}=\frac{FSR_{r}FSR_{s}}{\left|FSR_{r}-FSR_{s}\right|}\;$$

It has been demonstrated in the article [24] that when the FSR difference is large compared to the full-width at half-maximum of the resonance peaks of the individual resonators, digital biosensing can be realized. However, the spectrum cannot shift continuously, and the LOD is a fixed value that is equal to the FSR difference. This structure is improved by designing a small FSR difference compared to the full-width at half-maximum of the resonance peaks of the individual resonators. The periodic envelope signal can then be superposed onto the constituent peaks. In this case, the spectrum of the whole device can be continuously tracked, and the LOD is no longer determined by the FSR difference.

The optical material Ormocore utilized in our previous work is also adopted here. It has a large transparency window, including infrared and visible wavelengths, which makes it suitable for biosensing. All the designs are employed near 840 nm, where the water absorption loss is much lower than at the infrared wavelengths. The single-mode waveguide is designed with dimensions of 1×1 μm, in order to guarantee the single-mode condition. The coupling gap is designed to be 1 μm wide, as determined by the photolithography resolution of the contact mask aligner used in our laboratory. It can be seen from Formulas (1) and (2) that M is proportional to ${\mathrm{FSR}}_{r}$, and inversely proportional to the FSR difference. However, M can not grow without limit, as ${\mathrm{FSR}}_{t}\;$is conditioned by the available wavelength range of the measurement equipment. In our previous work, a single ring resonator with a radius of 220 µm was demonstrated to have good performance, and this radius was also adopted for the sensing microring here. Taking the measurement into consideration, at least two periods need to be confined within the available measurement wavelength range (10 nm here). After the simulation, the reference ring resonator was designed with a 210 µm radius.

## 3. Fabrication

The biosensor composed of two cascaded microrings was prepared on the polymer platform by using the Soft UV NIL technique, which is an improved version of the nanoimprint technique. As UV curing and a soft mold are employed, high temperatures and pressures are no longer needed, which significantly simplifies the imprinting process. A UV-curable polymer material, Ormocore, which has a large transparent window, was adopted to fabricate the device. The master mold in our work was fabricated on a negative photoresist, SU8-2 (MicroChemicals) by using UV lithography. A thickness of about 1 μm can be obtained by adjusting the spin-coating speed. A thermal reflow technique was also applied, as in our previous work, to improve the roughness of the waveguide sidewalls. For Soft UV NIL, polydimethylsiloxane (PDMS) was widely used to prepare the soft mold, due to its unique features such as backbone structure, a high degree of toughness, and large elongation. However, its surface energy (25 mN/m) is not small enough, which usually brings damage to the master mold, and influences its recycling. A novel material, polytetrafluoroethylene (PFPE), has been demonstrated as an ideal option for the soft mold in recent years [27]. It has many unique advantages, such as a low surface energy, an ideal elastic modulus, stability at an elevated temperature, and resistance to most chemicals. The fabrication process of the PFPE soft mold is illustrated in Figure 3a–c.

First, the PFPE liquid was obtained by mixing Irgacure 2022 photoinitiator (BASF) and Fluorolink MD 700 (Solvay Solexis) at a ratio of 1:20, and then cast onto the prepared master mold. Afterwards, a piece of polystyrene foil was carefully placed on the top to improve the mechanical stability. A roller was employed to press and flatten the soft mold in order to ensure that the master mold was fully filled, and the pattern was exactly replicated. Then, the mold was UV-cured for 90 s at am intensity of 30 mW/cm^2^. After peeling off from the master mold, the PFPE soft mold was prepared. The nanoimprint process is illustrated in Figure 3d–f. A Si wafer with a 3 μm SiO_2_ layer on the top, which acts as the under-cladding layer of the biosensor, was adopted as the substrate. First, an adhesion layer (Ormoprime) is prepared on the carefully cleaned substrate, and then it is cured on the hotplate at 150 °C for 5 min. Before spin-coating, the Ormocore liquid was diluted by the maT thinner (Microresist Technology), in order to obtain the designed waveguide height (1 μm). A filtration was also needed, in order to remove the particles in the mixture. In the next step, the Ormocore layer was deposited by spin-coating at 3000 rpm for 30 s and then it was baked at 120 °C for 10 min. After that, the imprinting was performed by carefully placing the PFPE soft mold onto the sample. An extra flattening process was unnecessary, as the capillary force is enough for nanoimprinting. Subsequently, UV-curing with an intensity of 30 mW/cm^2^ for 2 min was carried out in a nitrogen environment. Finally, a hard bake at 150 °C for 3 hr was applied after demolding the PFPE soft mold.

The fabricated two-cascaded-microrings biosensor was characterized by a Nova 600 Nanolab scanning electron microscope/focused ion beam (SEM/FIB) machine. As the machine combines a focused ion beam and an SEM together, cross-section imaging can be carried out easily. An SEM picture of a single microring is shown in Figure 4a, and it can be seen that the device can be prepared without defects in a large area. Figure 4b,c shows the top view and the cross-section of the coupling area of the microring resonator, respectively. It can be seen that the coupling gap was less than 1 μm, as was designed, and the waveguide was not completely rectangular. This is because the reflow technique made the waveguide deform slightly. Figure 4c also shows that the residual layer can be reduced to almost 0 nm. The residual layer is an important factor for microring biosensors, and it influences the coupling efficiency and bending loss. Much effort has been spent in our previous work to make this layer thin. In article [17], the residual layer below 100 nm has been demonstrated by using the minimum polymer squeezing method, which can significantly decrease the imprinting area. In article [10], PFPE soft molds based on different materials (MD 40 and MD 700) have been explored, and a residual layer as thin as 40 nm has been demonstrated by using MD 700. Based on our previous work, the proportion between the Ormocore and the maT thinner, which affects the viscosity of the mixture, has been explored here. The proportion between Ormocore and the maT thinner in our previous work was 1:2. Too much maT thinner makes the Ormocore layer so thin that the soft mold cannot be fully filled during imprinting. As a result, the waveguide cannot be replicated exactly. Too little maT results in thick residual layers. After optimization, an optimal proportion of 1:2.3 was obtained, to fabricate the biosensor, almost without a residual layer, which is shown in Figure 4c. Figure 4d is the oblique view of the fabricated waveguides, and a good degree of roughness can be observed. In general, this polymer-based cascaded-microrings biosensor can be fabricated to high quality by using the Soft UV NIL technique.

## 4. Measurement and Result Analysis

The fabricated two cascaded microrings biosensor is characterized by using a tunable laser and a power meter. A polarizer is placed between the laser and the biosensor chip, in order to obtain transverse-electric (TE) mode. The light is coupled in and out of the chip by using lensed fibers. A temperature-stabilized chuck is also applied during the measurement, in order to avoid the spectrum shift, due to temperature variations. The measurement is carried out at around 840 nm, and the measured transmission spectrum is given in Figure 5.

The measurement is applied in a wavelength range of 10 nm (from 833 nm to 843 nm), and two periods of the output envelope can be observed. In order to trace the spectral shift accurately, a two-step fitting method, including single-peak fitting and whole-envelope fitting was carried out. Since the peaks of the individual microring resonator can be well-approximated by a Lorentzian function, and the transmission of the two cascaded microring resonators can be described as a product of two microring resonator transmissions, each peak of the envelope spectrum in Figure 5 can be well-approximated by Formula (3), which is a product of two Lorentzian functions [24]:(3)$$T_{peak}\left(\lambda \right)=\frac{T_{max}{\left(\frac{FWHM_{ring}}{2}\right)}^{4}}{\left({\left(\frac{FWHM_{ring}}{2}\right)}^{2}+{\left(\lambda -\lambda_{m}-\frac{\Delta \lambda }{2}\right)}^{2}\right)\cdot \left({\left(\frac{FWHM_{ring}}{2}\right)}^{2}+{\left(\lambda -\lambda_{m}+\frac{\Delta \lambda }{2}\right)}^{2}\right)}\;$$
where $T_{max}$ is the maximum of the fitted peaks, $FWHM_{ring}$ is the full-width at half-maximum of the resonance peak of the individual microring resonator, and $\lambda_{m}$, Δλ are the mean and the difference between the two resonance wavelengths from two different microring resonators, respectively. Here, the full-width at half-maximum of the peaks is assumed to be identical for the individual microring resonators. This single peak fitting is employed on the highest constituent peaks of the first period of the measured spectrum, which is shown as a solid line in Figure 6.

After this single peak fitting, $\lambda_{m}$ and $T_{max}$ of each peak can be obtained, and will be utilized in the envelope fitting. The envelope consists of the maximum points of all the peaks, and it can be fitted to Formula (4):(4)$$T_{envelope}\left(\lambda \right)={\left(\frac{\sqrt{T_{max}}{\left(\frac{FWHM_{envelope}}{2\sqrt{\sqrt{2}-1}}\right)}^{2}}{{\left(\frac{FWHM_{envelope}}{2\sqrt{\sqrt{2}-1}}\right)}^{2}+{\left(\lambda -\lambda_{central}\right)}^{2}}\right)}^{2}\;$$
where $FWHM_{envelope}$ is the full-width at half-maximum of the fitted envelope, and $\lambda_{central}\;$is the central wavelength of the envelope. As $FWHM_{envelope}$ and $\lambda_{central}$ can be obtained by the two steps fitting, the spectral shift can be traced easily and accurately. After the fitting, the FSR of the whole envelope can be obtained as 4.6 nm, which is about 16 times larger than the FSR obtained from the single microring biosensor in our previous work [10]. The large FSR makes up for the disadvantages of the polymer-based microring resonators, which influence the measurement accuracy. Moreover, even though the bulk sensitivity has not been explored here, according to the Vernier effect, the sensitivity can also be improved by 16-fold. Deviations in the fabrication always affect the performance of the biosensor. According to Formula (2), the improved FSR is only related to the radius of the individual microring, which is replicated from the SU8-2 master mold. However, as the high quality contact mask which can be utilized repeatedly, is applied in the UV lithography, the influence on the radius from the fabrication deviation is negligible.

## 5. Conclusions

A biosensor based on two cascaded microrings has been demonstrated with a cost-effective polymer platform. The Vernier effect is utilized, to improve the FSR of the polymer-based single microring biosensor. The device is fabricated by using a UV-based soft imprint technique, with good quality. Ormocore, which has a large transparent window, is adopted. The proportion between Ormocore and the maT thinner is optimized, in order to minimize the residual layer, which is usually inevitable when using the nanoimprint technique. As a result, the device can be prepared almost without a residual layer. The fabricated biosensor is characterized by using a tunable laser and a power meter. The FSR of the whole envelope is 4.6 nm, which is about 16-fold the demonstrated value for the single-microring biosensor in our previous work (290 pm). Moreover, the bulk sensitivity also can be amplified by 16-fold, thanks to the Vernier effect. Finally, it is expected that the fabricated two-cascaded-microrings structure based on the Vernier effect has high potential for biosensing applications. 

## Figures and Tables

**Figure 1 sensors-19-00181-f001:**
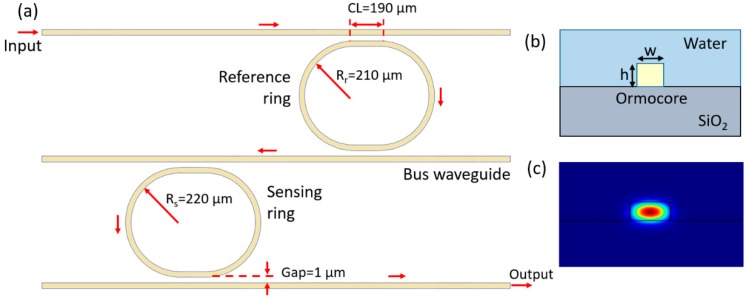
(**a**) The schematic configuration of the two-cascaded-microrings biosensor based on the Vernier effect. (**b**) Cross-section of the polymer-based waveguide. (**c**) Propagating mode in the polymer-based waveguide.

**Figure 2 sensors-19-00181-f002:**
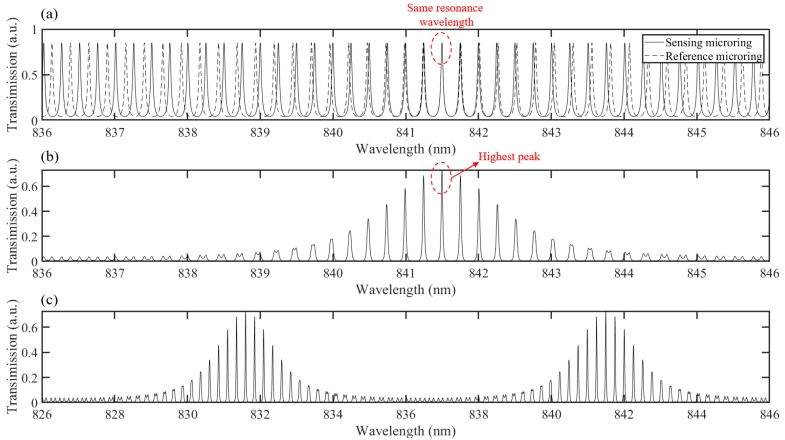
The schematic configuration of the Vernier principle. (**a**) Transmission spectrum of the sensing microring (solid line) and the reference microring (dashed line). (**b**) Total transmission spectrum of the two cascaded microrings. (**c**) A larger wavelength range including two periods, in order to show the enlarged FSR.

**Figure 3 sensors-19-00181-f003:**
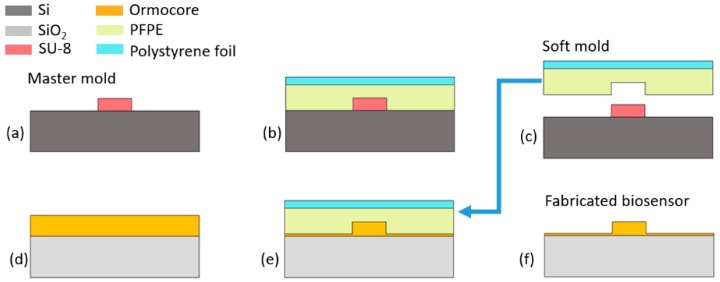
Fabrication process of the polytetrafluoroethylene (PFPE) soft mode (**a**–**c**) and UV-based soft imprinting process (**d**–**f**).

**Figure 4 sensors-19-00181-f004:**
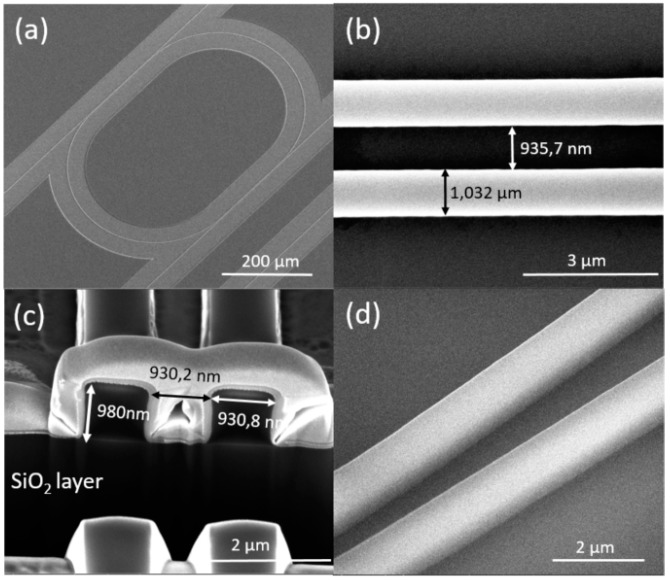
SEM pictures of different parts of the two cascaded microrings biosensor. (**a**) Top view of the single microring resonator. (**b**) Top view of the coupling section of the microring. (**c**) Cross-section of the coupling part. (**d**) The oblique view of the waveguide to show the roughness of the sidewalls.

**Figure 5 sensors-19-00181-f005:**
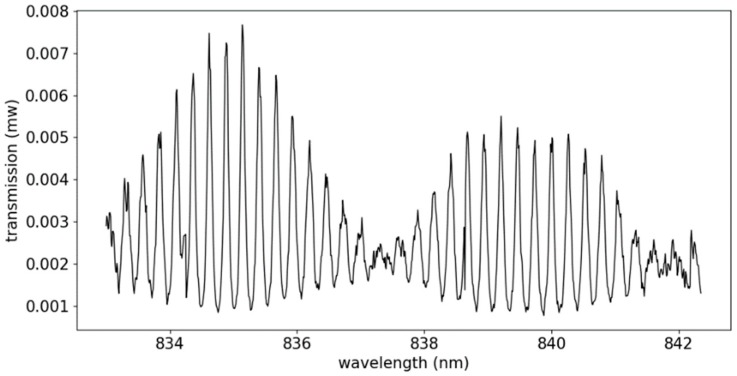
The measured transmission spectrum of the fabricated two-cascaded-microrings biosensor.

**Figure 6 sensors-19-00181-f006:**
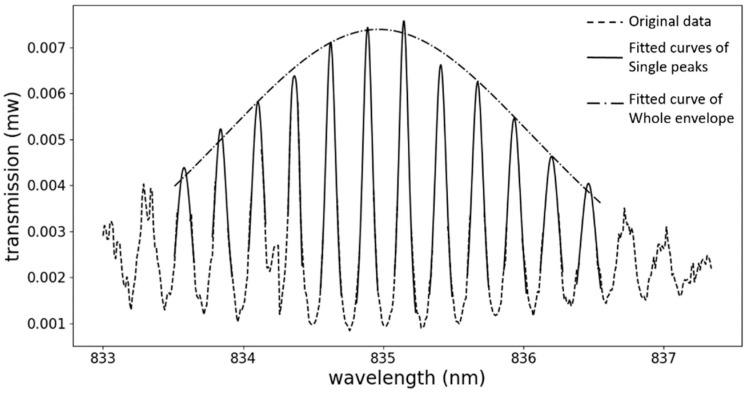
Illustration of the fitting procedure for the first period of the output envelope. The original data is shown, using dashed lines. The solid lines are the fitted curves of the single peaks, and the dash-dotted line is the fitted line of the whole envelope.

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
