# Peer review of "Cascaded-Microrings Biosensors Fabricated on a Polymer Platform"

_sensors, 2019, doi:10.3390/s19010181_

Round 1

Reviewer 1 Report

In this paper, the authors introduced single microring biosensors with good performance on the polymer platform. However, polymer waveguides have a small refractive index contrast which  leads to a large roundtrip length for the microring. As a result, polymer‐based microring biosensors usually have a small free spectral range (FSR) which is inversely proportional to the roundtrip   length and hampers the tracing of the spectrum shifting in the measurement. The idea behind this is interesting. However, I still have quite a number of concerns in this manuscript. There are times where there are not enough data to support the conclusions of the author. Please see some of the major concerns below.

1. The information of the microring structure is not enough (figure 1). The authors should give much more information about this. So the readers can get its reproducibility.  For example, authors can add the diameter values of the mirroring

2.  The authors should give much more information about the novelty of this paper, especially the effect of using this microring structure, which applications can be used this device?

3. The fabrication tolerance analysis, which can offer a good guide for the fabrication requirement, and the key parameters (R, FSR), need to be added in the results section.

4. More references need to be included in the introduction part to understand the applications of using multi micro-ring and silicon waveguide structures:

1.’Optical micro-multi-racetrack resonator filter based on SOI waveguides’, Photonics and Nanostructures – Fundamentals and Applications, 16, 2015 (16-23)

2. "Design of novel SOI 1x4 optical power splitter using seven horizontally slotted waveguides",Photonics and Nanostructures – Fundamentals and Applications, 25, 2017 (9-13)

3."Design of fiber-integrated tunable thermo-optic C-band filter based on coated silicon slab". J. of the European Optical Society-Rapid Publications, 13, 2017 (32)

5.  Much more discussion about the results should be given in this paper; especially the author needs to provide enough physicals mechanism analysis about the results. For example, can this device can cover the C-band or O-band for communication applications?

Reviewer 2 Report

The abstract needs a revision. An abstract is not a place to write the history of the project. I encourage authors to move the sentence to the introduction section and place before the last paragraph of the introduction section “ We have previously demonstrated single microring biosensors with good performance on the polymer platform. However, polymer waveguides have a small refractive index contrast which leads to a large roundtrip length for the microring. As a result, polymer‐based microring biosensors usually have a small free spectral range (FSR) which is inversely proportional to the roundtrip length and hampers the tracing of the spectrum shifting in the measurement.”

Page 1 Line 22-25: “Finally, the biosensor is characterized and the FSR is obtained as 4.6 nm which is 16 times that of the single microring biosensor demonstrated in our previous work.

Apples should be compared only with apples and not with the oranges. If the FSR of cascaded microring biosensor is 16 times larger than the FSR of the single microring biosensor, this should be clearly stated here.

Page 3 Line 89 Fig 1: Provide the dimension information of the waveguides. It's not clear from this figure how the structure is being used as a biosensor. Thoroughly revise it.

Page 3 Line 110: Magnification of the images are done by the optical lenses. However, the sensitivity of the biosensors is amplified using various methods as magneto-optic modulations, etc. Usually, magnification is reserved in optics studies. The author might consider replacing the term “magnify” by the word “amplify”.

Page 3 Line 110: If equations are borrowed from the textbook, this should be cited here.

Page 4 Line 137-138: Label the figure appropriately, increase the fonts where necessary, provide units in the y-axis, provide legends.

Page 4 Line 144: “Two biosensors” and “is” prepared? This is the most fundamental parts of English taught in elementary school. Authors should read their manuscript carefully and revise the sentence with the correct grammatical format as Two… are… prepared.

Page 5 Line 159: Label the fabrication steps appropriately. Provide the dimension information

Page 6 Fig 5: Provide additional information about the characteristics in the inset, label the main peaks and corresponding wavelengths, if possible.

Page 6 Line 218: Authors should cite the equations even if it was borrowed from author’s previously published literature.

Page 7 Line 227, Fig 6: Label the curve appropriately instead of describing it in the caption. Remember one picture is worth a thousand word if drawn appropriately.

Conclusion section:

Please remember, we are in the conclusion section. This section should not be used to tell the story again. We already passed the storytelling section.

Page 7 Line 246: Through the garbage somewhere else. The introduction section is already too large, but this information should not be sitting here. “A lot of work has been devoted to polymer‐based microring biosensors due to their advantages and many good results have been demonstrated. However, polymer waveguides have a low refractive index contrast, which normally results in small FSRs of the microring resonators. In previous work, we obtained an FSR of 290 pm which is so small that it influences the tracing of the spectral shift.”

Line 251: Line 256: This is not an essay writing competition. Additional junk is sitting here: Move it to elsewhere: “The FSR, as well as the sensitivity, can be magnified by the Vernier effect generated by the two cascaded microrings. The UV‐based soft imprint technique as an improved version of the nanoimprint technique is applied to fabricate the device. As high pressure and temperature are unnecessary, this technique is much simpler and cost‐effective. However, an additional requirement is that the utilized polymer must be UV curable.”

Only present the results conclusive to this work.

For reference on MZI and plasmonic-based biosensors, refer the papers as:

1)"Improved magneto-optic surface plasmon resonance biosensors." Photonics. Vol. 5. No. 3. Multidisciplinary Digital Publishing Institute, 2018. Photonics 20185(3), 15; https://doi.org/10.3390/photonics5030015

2) "Compact Si-based asymmetric MZI waveguide on SOI as a thermo-optical switch." Optics Communications 410 (2018): 947-955. https://doi.org/10.1016/j.optcom.2017.10.007

Round 2

Reviewer 1 Report

The modified paper can be publish

Author Response

Thank you for your kind suggestion!

Reviewer 2 Report

Authors have revised the abstract, have provided cross-sectional information of the waveguide and inserted key parameters in Fig 1. They have explained the sensing principle, done grammatical corrections, added relevant citations for borrowed equations. Authors have revised Figures have labelled curves in the Figures. The conclusion section has been revised and additional references relevant to the manuscript has been added.

Authors have briefly mentioned the effect of waveguide size during fabrication. Wondering if the wavelength of the laser was temperature sensitive? If the measurement results are temperature sensitive, authors should add this information in the measurement section. The paper should be ready for publication after the careful checkup. 
